# Ectopic overexpression of *Plasmodium falciparum* DNA-/RNA-binding Alba proteins misregulates virulence gene homeostasis during asexual blood development

Dimple Acharya,[1,2] Anitha Nagaraj Bavikatte,[2] Vishnu Vinayak Ashok,[2] Shubhada R. Hegde,[2] Cameron Ross Macpherson,[3,4,5] Artur Scherf,[3,4,5] Shruthi Sridhar Vembar[2]

**ABSTRACT**  Alba domain-containing proteins are ubiquitously found in archaea and eukaryotes. By binding to either DNA, RNA, or DNA:RNA hybrids, these proteins function in genome stabilization, chromatin organization, gene regulation, and/or translational modulation. In the malaria parasite *Plasmodium falciparum*, six Alba domain proteins PfAlba1–6 have been described, of which PfAlba1 has emerged as a "master regulator" of translation during parasite intra-erythrocytic development (IED). Given that a tight control of gene expression is especially important during IED, when malaria pathogenesis manifests, in this study, we focus on three other *P. falciparum* Albas, PfAlba2–4. Because genetic manipulation of the genomic loci of PfAlba2–4 was unsuccessful, we overexpressed each of these proteins from an episome under a strong constitutive promoter. We observed that PfAlba2 or PfAlba3 overexpression strongly reduced parasite growth and impacted IED stage transitions. In contrast, elevated levels of PfAlba4 were well-tolerated by the parasite. In keeping with this, differential gene expression analysis using RNA-seq of PfAlba2 or PfAlba3 overexpressing strains revealed a significant misregulation of mRNAs encoding virulence factors, such as those related to erythrocyte invasion; a general repression of *var* gene expression was also apparent. PfAlba4 overexpression, on the other hand, did not significantly perturb the steady-state transcriptome of IED stages and appeared to enhance *var* mRNA levels. Moreover, distinct sets of genes were targeted by each PfAlba for regulation. Taken together, this study highlights the nonredundant roles of PfAlba proteins in the *P. falciparum* IED, emphasizing their importance in subtelomeric chromatin biology and RNA regulation.

**IMPORTANCE** The malaria parasite *Plasmodium falciparum* tightly controls the expression of its genes at the epigenetic, transcriptional, post-transcriptional, and translational levels to synthesize essential proteins, including virulence factors, in a timely and spatially coordinated manner. A family of six proteins implicated in this process is called PfAlba, characterized by the presence of the DNA-, RNA- or DNA:RNA hybrid-binding Alba domain. To better understand the cellular pathways regulated by this protein family, we overexpressed three PfAlbas during *P. falciparum* intra-erythrocytic growth and found that high levels of PfAlba2 and PfAlba3 were detrimental to parasite development. This was accompanied by significant changes in the parasite's transcriptome, either with regards to mRNA steady-state levels or expression timing. PfAlba4 overexpression, on the other hand, was well-tolerated by the parasite. Overall, our results delineate specific pathways targeted by individual PfAlbas for regulation and link PfAlba2/PfAlba3 to mutually exclusive expression of the virulence-promoting surface antigen PfEMP1.

**Peer Reviewer** Francesca Florini, Weill Cornell Medicine, New York, New York, USA

Address correspondence to Shruthi Sridhar Vembar, ssvembar@ibab.ac.in.

The authors declare no conflict of interest.

This work was presented at the Molecular Parasitology Meeting (XXXI) 2020, organized by Marine Biology Laboratory, Woods Hole, Massachusetts, USA; the National Conference on CRISPR/Cas: From Biology to Technology (iCRISPR 2021), organized by IBAB, Bengaluru, India, and SRM-AP, Amaravati, India; and the 32nd National Congress of Parasitology 2024, organized by the Indian Society of Parasitology at IISER-Pune, India.

**KEYWORDS** gene regulation, *Plasmodium falciparum*, transcriptomics, Alba domain, DNA-binding proteins, RNA-binding proteins, RNA-seq

*P*lasmodium falciparum, a unicellular Apicomplexan parasite, which causes the deadliest type of human malaria, develops asexually over a 48-hour time period within human erythrocytes to mount a successful infection. Of the 5,300 protein-coding genes that are contained within its 23-Mb AT-rich genome, *P. falciparum* expresses nearly 85% during asexual blood growth. Similar to other eukaryotes, gene regulation in *P. falciparum* takes place at the epigenetic, transcriptional, post-transcriptional, co-translational, and post-translational levels (1, 2). Early studies of the asexual stage transcriptome and proteome suggested that many genes encoding virulence factors exhibit either just-in-time transcription (*i.e.*, mRNA is made and translated when the protein function is required) or just-in-time translation (*i.e.*, the mRNA is made and stored in a translationally repressed state until the protein function is required) (3–6). The former is regulated at the epigenetic and transcriptional levels and is relatively well-studied, while the latter is regulated at the post-transcriptional and co-translational levels, and not as well understood (7–12). Given that, in the past decade, more than 1,000 RNA-regulatory proteins with distinct RNA-binding domains (13–15) have been identified as part of the *P. falciparum* proteome, there is burgeoning interest in understanding their role in post-transcriptional and co-translational gene regulation, in turn shedding light on the just-in-time translation phenomenon.

A family of RNA-binding proteins (RBPs) that was first identified in the mouse malaria parasite *P. berghei* as part of a translational repressor complex (16) and subsequently in *P. falciparum* (17) is the Alba (acetylation lowers binding affinity) family. The Alba domain is ~90 amino acids in length and was initially discovered in archaea as an architectural protein domain involved in genome organization (18). Subsequent studies suggested that the Alba domain evolved from the ancient IF3-C RNA-binding fold (19, 20), with several reports confirming RNA and/or DNA regulatory activities of this domain in a variety of organisms. For example, in *Toxoplasma gondii,* a parasite that is closely related to *P. falciparum*, TgAlba1 and TgAlba2 play a role in translational regulation and differentiation, most likely by binding to specific RNA targets (21). This is also the case in the Discoban parasite *Trypanosoma brucei,* where four Alba domain proteins TbAlba1–4 have been demonstrated to have cytoplasmic RNA-binding activity, implying that they are either part of the translation machinery or that they are involved in translational control (22, 23). In another Discoban parasite *Leishmania infantum*, LiAba1 and LiAlba2 associate with ribosomal subunits, translation factors, and other RBPs, again implying a role in post-transcriptional and/or co-translational regulation (24). In *Arabidopsis thaliana*, which encodes for six Alba domain proteins, AtALBA4, AtALBA5, and AtALBA6 play a role in thermotolerance by stabilizing the mRNAs of heat stress transcription factors in stress granules and processing bodies (25). Additionally, a report in 2021 implicated AtALBA1 and AtALBA2 in the regulation of genome stability through binding to DNA:RNA hybrids within R-loop structures (26). In *Saccharomyces cerevisiae,* Alba domain-containing Pop6 and Pop7, which are orthologous to Alba family proteins Rpp20 and Rpp25 of humans (27), form integral components of the RNAse P and RNAse MRP holoenzymes (28). Interestingly, recent studies have shown that *S. cerevisiae* Pop6 and Pop7 also associate with telomeric complexes (29), suggesting that they may also play a role in telomere end biology either through protein–protein, protein–RNA, or protein–DNA interactions. Similarly, archaeal Alba domain proteins, in addition to their role in DNA condensation and chromatin organization (19, 20, 30), have been implicated as RNA chaperones (31), stabilizers of double-stranded RNA (32–34), and potentially in translational control (34). Overall, Alba domain-containing proteins have gained increasing interest in recent years due to their widespread distribution, presence in multiple copies, functional crosstalk, differential binding affinity to nucleic acids, and nucleocytoplasmic shuttling.

In *P. falciparum*, we first identified the PfAlbas (Fig. 1A) as part of a protein complex that associates with sub-telomeric DNA repeat sequences and subsequently showed that four of the six PfAlbas, PfAlba1–4, bind to RNA and DNA *in vitro* (17). PfAlba5 and PfAlba6 were identified later using a bioinformatics approach (15), and to date, their functional validation lags behind, although a recent study showed that PfAlba6 is a DNA-targeted exonuclease, which may regulate the parasite stress response during asexual blood stage development (35). Additional studies showed that, in ring stages, all four PfAlbas directly or indirectly bind to the intron of *var* genes (36, 37), where *var* is a 60-member virulence gene family located either at the sub-telomeric regions of chromosomes 1 through 13 or within central regions of chromosomes 4, 6, 7, 8, and 12, whose mutually exclusive expression is epigenetically regulated. We also reported that PfAlbas exhibit temporally regulated intracellular localization during asexual growth: for instance, PfAlba1, PfAlba2, and PfAlba3 are nuclear during early stages of asexual growth (0–24 hours) and distribute as punctate foci in the cytoplasm of late-stage parasites (24–48 hours). The latter observation is supported by the abundance of PfAlba1–4 in the mRNA-bound proteome and polysome-associated fractions of late asexual stage *P. falciparum* parasites, implying a critical function for these proteins in regulating mRNA localization, stability, translation, and/or degradation (13). Following this, our detailed characterization of PfAlba1 revealed its involvement in regulating the translational timing of mRNAs encoding erythrocyte invasion genes (38), suggesting that PfAlbas may be master regulators of mRNA homeostasis (11, 13, 39). More recently, PfAlba2 was identified as an interactor the GC-rich noncoding RNA (ncRNA) RUF6, which is an activator of *var* expression (40), and PfAlba3 was shown to possess endonuclease activity *in vitro* (41). Nonetheless, beyond PfAlba1, there have been no *in vivo* Alba-focused studies, and we sought to fill this knowledge gap.

Our consistent efforts to modify the genomic loci of PfAlba2–4, either to knockout the gene or tag the gene for conditional knockdown, have not been successful, even with CRISPR/Cas9 gene editing. This is further supported by genome-wide transposon–mutagenesis screens that categorized PfAlba2–4 to be essential for asexual growth (42). Therefore, we overexpressed these proteins with a C-terminal Ty1 epitope tag from an episome to better understand their gene regulatory role. After confirming the transgenic parasites, we analyzed their intra-erythrocytic development (IED) and performed transcriptomic analysis using RNA-seq. We observed that overexpression of PfAlba2-Ty1 or PfAlba3-Ty1, but not PfAlba4-Ty1, inhibited parasite growth possibly by misregulating the timing of different developmental processes. Moreover, PfAlba2 and PfAlba3 overexpression strongly perturbed the steady-state IED transcriptome and globally repressed the *var* multigene family, with PfAlba3-overexpressing parasites showing the most dramatic transcriptional changes (*i.e.,* more than 30% of the parasite's genes are up- or downregulated). Overall, this study strongly implicates PfAlba2 and PfAlba3 as central players in *P. falciparum* IED gene regulation, with a disruption in their homeostasis affecting parasite growth and virulence gene expression.

## MATERIALS AND METHODS

### Parasite culture and transfection

Asexual blood stages of the *P. falciparum* laboratory strain 3D7 and its transfectants were cultured as described previously (17). Giemsa staining of parasites to determine parasitemia and parasite developmental age was carried out as per established protocols (43). Transfection was performed with 3D7 ring stage parasites (44) with 50 or 100 µg of the plasmid constructs. Drugs used were blasticidin-S (BS; Invivogen) at 2.5 or 5 µg/mL. Transfectant parasites appeared after 20–25 days of drug pressure.

### Generation of plasmid constructs

The pLN-Ty1C and pPfAlba4-Ty1C have been previously reported (17). The pPfAlba2-Ty1C and pPfAlba3-Ty1C constructs were generated by inserting the coding region of

*ALBA2* and *ALBA3*, respectively, without the STOP codon, upstream of and in-frame with the Ty1 epitope in pLN-Ty1C.

## Antibodies and Western blotting

Western blotting was performed with rabbit anti-Ty1 antibodies (Genscript), mouse BB2 monoclonal anti-Ty1 antibodies (17), rabbit anti-PfAlba2 antisera (17), mouse anti-PfAlba3 antisera (17), and rabbit anti-PfAlba4 antisera (17). PfHsp70 or PfAldolase (antibodies obtained from AbCam) was used as a loading control, and all signals were detected with HRP-conjugated secondary antibodies. Images were captured using a BioRad ChemiDoc system and signals quantified using ImageJ (45).

## Immunofluorescence microscopy

Asynchronous cultures of 3D7 + empty vector, 3D7 + PfAlba2-Ty1, 3D7 + PfAlba3-Ty1, or 3D7 + PfAlba4-Ty1 parasites growing in the presence of 5 µg/mL of blasticidin-S were prepared for immunofluorescence assays as previously described (17) using rabbit anti-Ty1 antibodies (Genscript), rabbit anti-PfAlba2 antibodies, and rabbit anti-PfAlba4 antibodies. AlexaFlour564-conjugated goat anti-rabbit antibodies were used for detection. Labeled parasites were deposited on microscopy coverslips and mounted onto slides in Fluoroshield Anti-Fading Mounting media supplemented with DAPI (Sigma-Aldrich). Images were captured using a Zeiss LSM 880 Airyscan microscope and analyzed with the Zeiss zen (v3.1) software. Calculation of co-localization of PfAlba-Ty1 and DAPI signals was performed using ImageJ with Just Another colocalization (JAcoP) plugin. The Pearson's coefficient and Mander's coefficient, after using thresholds, are reported here.

## Flow cytometry

To measure the growth rate of transfectant parasites, synchronous cultures of ring stages were diluted to 0.2% parasitemia in 200 µL RPMI complete medium at 4% hematocrit. At 0 hour, 24 hours, 48 hours, 72 hours, 96 hours, 120 hours, and 144 hours, 5 µL of the culture was stained in 95 µL of D-PBS (Gibco) supplemented with 2 × Sybr Green I (Ozyme; stock = 10,000 × ) for 30 minutes at room temperature, diluted 20-fold in D-PBS (final volume = 200 µL), and the SYBR Green fluorescence measured in a Guava easyCyte Flow Cytometer (EMD Millipore). We counted 10,000 events in duplicate or triplicate to establish an accurate parasitemia value for each culture. Data were captured and analyzed using the InCyte software (EMD Millipore). To analyze changes in DNA content and lifecycle stages during the 48-hour IED cycle of 3D7 + empty vector, 3D7 + PfAlba2-Ty1, 3D7 + PfAlba3-Ty1, or 3D7 + PfAlba4-Ty1 parasites, synchronous ring stages growing in RPMI complete medium supplemented with 5 µg/mL BS were diluted to a parasitemia of 0.5% at 4% hematocrit: this was considered to be the "0 h" time point. The parasitemia of the culture was subsequently analyzed every 3 hours for up to 70 hours by flow cytometry, as described above. To accurately demarcate ring, trophozoite, and schizont stages, data were analyzed using FlowJo v.9 software (FlowJo, LLC).

## Preparation of strand-specific RNA-seq libraries and sequencing

For transcriptomic analysis, total RNA from synchronized 3D7, 3D7 + empty vector, 3D7 + PfAlba2-Ty1, 3D7 + PfAlba3-Ty1, or 3D7 + PfAlba4-Ty1 parasites at the ring (8–10 hour) and trophozoite (28–30 hour) stages was prepared using the miRNeasy mini kit (Qiagen) according to the manufacturer's instructions; to reduce human RNA contamination, the parasites were grown in human blood free of white blood cells, in RPMI complete medium supplemented with 5 µg/mL BS. Next, ~10–15 µg total RNA was treated with DNase using the Turbo DNA-free kit (Thermo Fisher Scientific), poly(A)-enriched using the Dynabeads mRNA purification kit (Thermo Fisher Scientific), and used for strand-specific RNA-seq library preparation. A minimum of two biological replicates were analyzed for each experimental and control sample. Strand-specific Illumina sequencing libraries

were prepared as described (46) using 14–20 cycles of library amplification. The resulting preparations were sequenced using a 75 or 100 nucleotide single-end run on a HiSeq 2000 (Illumina). The Illumina image files were converted to fastq format using bcl2fastq (www.illumina.com). The total number of sequenced reads for each fastq file generated in this study is summarized in Table S1. Note that, for the 3D7 strain, only one of the two ring stage replicates sequenced, and for the 3D7 + PfAlba3-Ty1 strain, only two of the four ring stage replicates sequenced, had more than one million read counts (data not shown) and were retained for downstream analysis.

## Analysis of RNA-seq data

The quality of raw reads in the fastq files was checked using FASTQC (https://www.bioin-formatics.babraham.ac.uk/projects/fastqc/). To retain high-quality reads for downstream analysis, adapter and low-quality reads were trimmed by Trim galore v0.6.4_dev (47). Next, the *P. falciparum* 3D7 reference genome (v3, release 46) was downloaded from PlasmoDB (https://plasmodb.org/; 48) and indexed using the STAR aligner (2.7.3 a) (49) following which the trimmed fastq files were aligned to the reference using STAR. The resulting sam files were converted to bam files using Samtools v1.10 (50), and reads with a mapping quality score of 40 or higher were retained. Using the bam files, raw read counts for each *P. falciparum* gene were calculated using HTSeq 0.11.1 (51; Table S2), and Transcripts per million (TPM) normalization was performed using R script. For the TPM normalized data, Pearson Correlation Coefficient (PCC) values were calculated using R (1.0.12), and pheatmap (52) was used for PCC plot generation. Principal Component Analysis (PCA) was performed using TPM normalized data in R, and eigenvectors and eigenvalues were calculated (1.0.12). Maximum likelihood-based statistical age analysis of the read count data was performed as previously described (38, 53), and an R script was used for plotting the results in ggplot2 (3.3.6).

The raw read counts were subsequently used for differential expression analysis in DESeq2 1.27.11 (54). Genes with $|\log_2(\text{Fold Change})|$ cut-off of $>= 1.5$ and false discovery rate-corrected *P*-value of $<= 0.05$ were selected for Gene Ontology (GO) analysis in PlasmoDB (https://plasmodb.org). Intersection analysis was performed using Lucidchart (https://www.lucidchart.com). RPKM values for *var, rifin,* and RUF6 genes were calculated using Microsoft Excel. ggplot2 (3.3.6) was used to generate the PCA plots, while GraphPad Prism or Microsoft Excel was used to generate other plots. Weighted Gene Co-expression Network Analysis (WGCNA) was done by using WGCNA library (1.72.1) (55).

## RESULTS

### Generation of *P. falciparum* transgenic lines ectopically expressing C-terminally Ty1-tagged PfAlba2, PfAlba3, or PfAlba4

In a previous study, we sought to delineate the function of PfAlba1 by overexpressing the protein from an episome with a C-terminal Ty1 epitope tag (38). Similarly, we constructed pLN-Ty1-based constructs for PfAlba2, PfAlba3, or PfAlba4, in which the expression of C-terminally Ty1-tagged versions of these proteins is driven by the promoter of the *P. falciparum* calmodulin gene (*PF3D7_1434200; P_cam*). Post-transfection and drug selection, we first confirmed the presence of the plasmid in the transgenic lines using PCR genotyping (Fig. 1B). Next, using Western blotting, we detected PfAlba2-Ty1, PfAlba3-Ty1, or PfAlba4-Ty1 proteins at the expected size with anti-Ty1 antibodies (Fig. 1C). We also analyzed the localization of the tagged proteins in ring, trophozoite, and schizont stages using immunofluorescence assays and found that PfAlba2-Ty1 primarily localized to the cytoplasm in all stages, with partial nuclear localization in ring stages alone (Fig. 1D; Fig. S1B). In contrast, the PfAlba3-Ty1 signal was found to be both nuclear and cytoplasmic in all stages (Fig. 1D; Fig. S1B), while PfAlba4-Ty1 appeared to be predominantly nuclear in ring stages, with the signal spreading to the cytoplasm in later stages (Fig. 1D; Fig. S1B). Staining of the 3D7 + PfAlba2-Ty1 and 3D7 + PfAlba4-Ty1

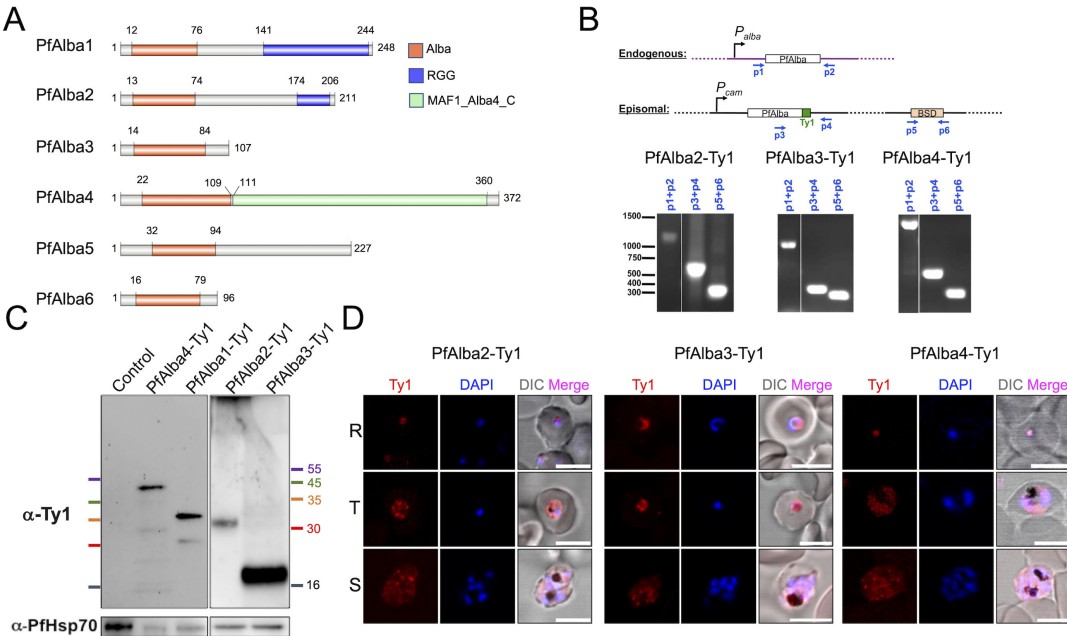

**FIG 1** C-terminally tagged PfAlba2-Ty1, PfAlba3-Ty1, and PfAlba4-Ty1 were successfully expressed from an episome in *P. falciparum* asexual blood stages. (A) Schematic representation of the domain organization of the six Alba proteins of *P. falciparum*. (B) 3D7 parasites transfected with the pLN-PfAlba2-Ty1, pLN-PfAlba3-Ty1, or pLN-PfAlba4-Ty1 plasmids were grown in the presence of 5 mg/mL of blasticidin-S (BS), their genomic DNA harvested, and used with the indicated primers in PCRs to confirm the uptake of the episome. Primer pairs targeted either the genomic *Alba* locus (*p1 +p2*) or different regions of the episome (*p3 +p4* and *p5 +p6*) as shown in the scheme (*top panel*), which is not drawn to scale. The DNA size marker corresponds to the GeneRuler 1 kb Plus DNA Ladder (Fermentas, Thermo Fisher Scientific). (C) PfAlba2-Ty1, PfAlba3-Ty1, or PfAlba4-Ty1 proteins were detected in protein lysates prepared from the indicated transfectant lines by Western blotting with mouse anti-Ty1 antibodies. PfHsp70 served as a loading control. (D) Immunofluorescence assays (IFAs) were used to determine the localization of PfAlba2-Ty1, PfAlba3-Ty1, and PfAlba4-Ty1 in ring (R), trophozoite (T), and schizont (S) stages of the indicated transfectant lines. Antibodies used included rabbit anti-Ty1 (red). Nuclei were labeled with DAPI (blue). Scale bar represents 5 µM. All of the cultures used for Western blotting and IFAs contained 5 µg/mL of blasticidin-S.

transfectants with anti-PfAlba2 and anti-PfAlba4 antibodies, respectively, recapitulated these results (Fig. S1C and D).

## Ectopic expression of PfAlba2-Ty1 or PfAlba3-Ty1, but not PfAlba4-Ty1, inhibits intra-erythrocytic growth of *P. falciparum*

To understand the importance of PfAlba protein homeostasis to *P. falciparum* IED and survival, we modulated the levels of the Ty1-tagged PfAlba proteins by increasing blasticidin-S (BS) drug concentration in the growth media. 3D7 + PfAlba2-Ty1, 3D7 + PfAlba3-Ty1, or 3D7 + PfAlba4-Ty1 ring-stage parasites that were growing in media containing 2.5 µg/mL BS were transferred to media containing 0, 5, 10, or 20 µg/mL BS, and their growth was monitored over a 5-day period using SYBR Green staining and flow cytometry. As a control, we used 3D7 parasites transfected with the empty pLN-Ty1 vector. We observed that, after two replication cycles, when compared to the control, the parasitemia of 3D7 + PfAlba2-Ty1 and 3D7 + PfAlba3-Ty1 cultures reduced significantly at 10 and 20 µg/mL BS relative to the control, while there was no reduction in the parasitemia of 3D7 + PfAlba4-Ty1 (Fig. 2A; red arrowhead indicates parasites grown at 10 µg/mL BS). Even at 0 and 5 µg/mL BS and within a single replication cycle, the 3D7 + PfAlba2-Ty1 and 3D7 + PfAlba3-Ty1 transfectants showed a lower multiplication rate compared with empty vector transfectants (Fig. 2A), indicating that optimal levels of PfAlba2-Ty1 and PfAlba3-Ty1 levels are required for parasite growth under laboratory conditions; of note, the expression levels of the tagged proteins were similar at 0 and 5 µg/mL BS (Fig. S2A and B). Multiple attempts to detect changes in PfAlba levels in response to increasing BS concentration (>5 µg/mL BS) in the growth media by Western

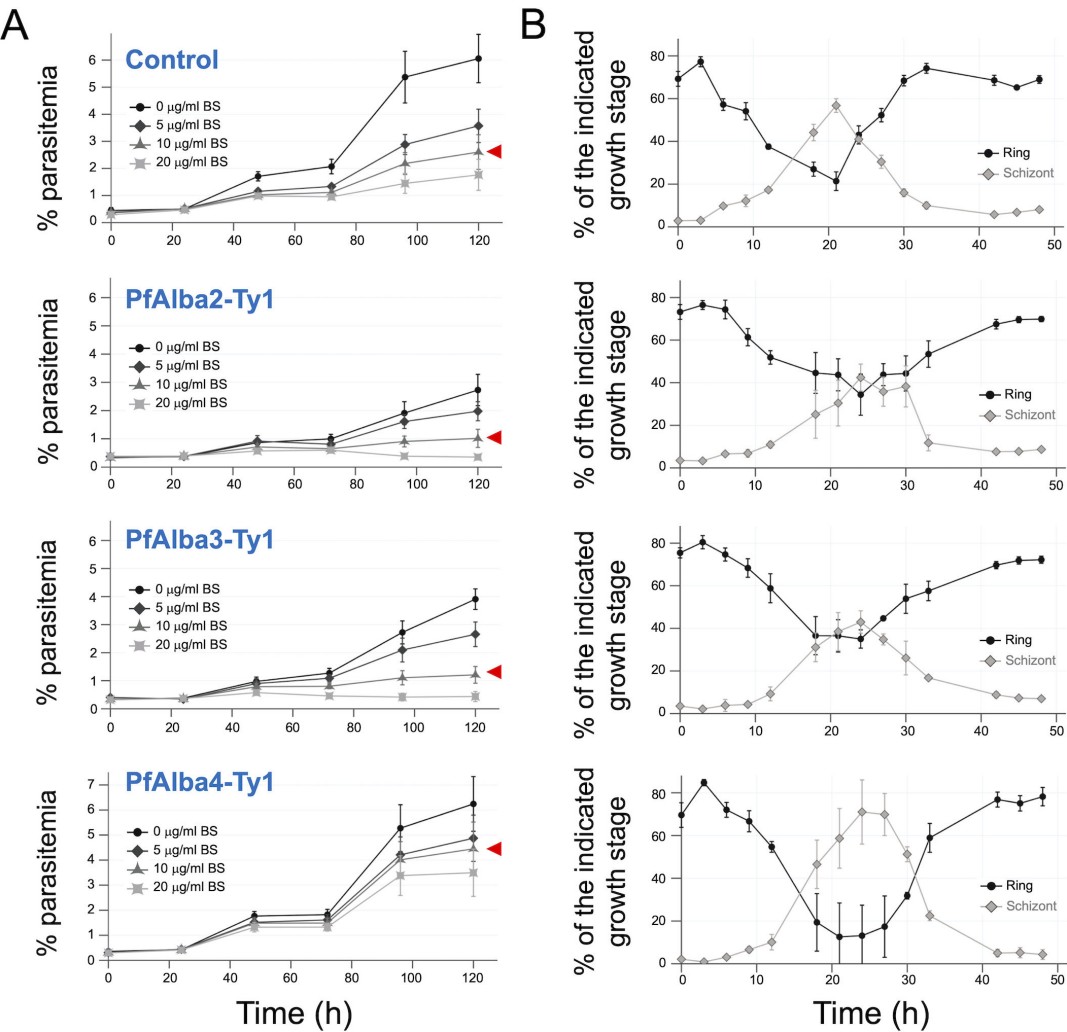

**FIG 2** Overexpression of PfAlba2 or PfAlba3 adversely affects intra-erythrocytic growth of *P. falciparum*. (A) The growth of 3D7 + empty vector, 3D7 + PfAlba2-Ty1, 3D7 + PfAlba3-Ty1, and 3D7 + PfAlba4-Ty1 parasites was measured by flow cytometry for 5 days in the presence of the indicated concentrations of BS (in µg/mL). The *y-axis* denotes the percentage parasitemia at each time point. Data represent the means of a minimum of three independent experiments ± SEM (error bars). Red arrows indicate growth profiles of the strains at 10 µg/mL BS concentration. (B) The ~48-h IED cycle of the Alba-overexpressing lines was monitored by flow cytometry. The *y-axis* denotes the percentage of ring or schizont stages at each time point. Data represent the means of a minimum of three independent experiments ± SEM (error bars).

blotting with anti-Ty1 antibodies were unsuccessful for the 3D7 + PfAlba2-Ty1 and 3D7 + PfAlba3-Ty1 strains, possibly because of low parasite numbers at higher BS concentration (data not shown). This was not the case with the 3D7 + PfAlba4-Ty1 strain, where we observed a dose response of PfAlba4-Ty1 protein levels to changes in BS concentration (Fig. S2C and D), as we had previously reported for PfAlba1-Ty1 (38).

We next measured the cell cycle time and stage conversion of 3D7 + PfAlba2-Ty1, 3D7 + PfAlba3-Ty1, or 3D7 + PfAlba4-Ty1 parasites growing at a concentration of 5 µg/mL BS by SYBR Green I staining and flow cytometry. We observed a marked difference in the progression of ring to late stages between the control and 3D7 + PfAlba2-Ty1 or 3D7 + PfAlba3-Ty1 transfectants, but not 3D7 + PfAlba4-Ty1 (Fig. 2B; Fig. S3), implying that the growth defects of 3D7 + PfAlba2-Ty1 and 3D7 + PfAlba3-Ty1 parasites are likely due to a dysregulation of cell cycle timing, although we did not observe a gross change in parasite morphology in Giemsa-stained blood smears (Fig. S4). Taken together, our results indicate that excess amounts of PfAlba2 and PfAlba3 reduce *P.*

*falciparum* intra-erythrocytic growth in a dose-dependent manner and by altering cell cycle progression.

## Transcriptomic analysis of PfAlba2-Ty1, PfAlba3-Ty1, and PfAlba4-Ty1 transfectants

Given that the PfAlbas are DNA-/RNA-binding proteins, we examined the effects of ectopic PfAlba2-Ty1, PfAlba3-Ty1, or PfAlba4-Ty1 expression on the *P. falciparum* blood stage transcriptome. We performed RNA-seq analysis of two intra-erythrocytic stages within the same replication cycle, ring (8–10 hours post-invasion (hpi)) and trophozoite (28–30 hpi) stages, of control and 3D7 + PfAlba2-Ty1, 3D7 + PfAlba3-Ty1 or 3D7 + PfAlba4-Ty1 transfectants, growing at a concentration of 5 µg/mL BS. For each time point, two RNA-seq biological replicates were generated for 3D7 + PfAlba2-Ty1, 3D7 + PfAlba3-Ty1, or 3D7 + PfAlba4-Ty1 transfectants, while for the control, three and four biological replicates were generated, respectively, for the ring and trophozoite stages (also see Materials and Methods). Fig. 3A summarizes the data analysis pipeline, starting with quality control of the transcriptomic data sets.

First, to determine the reproducibility of our RNA-seq experiment, we estimated the Pearson correlation coefficient (PCC) of all the samples in a pairwise manner, based on Transcripts Per Million (TPM)-normalized read counts, and found that samples of a specific stage had higher pairwise PCC values and clustered together (Fig. 3B). The only exception was a ring stage biological replicate R2 of 3D7 + PfAlba3-Ty1, which clustered with trophozoite stage samples despite showing a high correlation to ring stage replicate R1 of 3D7 + PfAlba3-Ty1; we, therefore, retained this sample as a ring replicate for downstream analysis (also see *Materials and Methods*). Subsequently, we explored the global differences between the transcriptomes of the control and Alba-Ty1 transfectants using PCA and observed that ring and trophozoite stage samples differentiated along PC1, while the controls and 3D7 + PfAlba4-Ty1 samples were distinguished from the 3D7 + PfAlba2-Ty1 and 3D7 + PfAlba3-Ty1 samples along PC2 (Fig. 3C). This suggested that PfAlba4-Ty1 ectopic expression impacted the *P. falciparum* blood stage transcriptome to a lesser extent than PfAlba2-Ty1 or PfAlba3-Ty1 expression. We also analyzed the transcript levels of PfAlba1, PfAlba2, PfAlba3, and PfAlba4 in all RNA-seq samples and noticed that the steady-state expression of each PfAlba appeared to be similar between the overexpressor strains and the controls, the only exception being PfAlba4 (Fig. S5). One explanation for this could be that there may be a negative feedback loop for self-regulation by PfAlba2 and PfAlba3, which remains to be explored.

Lastly, we used a maximum likelihood-based statistical analysis method (53) to estimate the developmental age of our PfAlba2-Ty1, PfAlba3-Ty1, and PfAlba4-Ty1 transcriptomic data sets. This analysis separates true differential expression from cell cycle-dependent, temporal changes in expression patterns and is an important measure of the quality of our RNA-seq replicates for technical and biological variability. We observed that the transcriptome of controls, 3D7 + PfAlba2-Ty1, 3D7 + PfAlba3-Ty1, and 3D7 + PfAlba4-Ty1 ring stages mapped to the same early time point of the parasite life cycle (8–10 hpi) for all replicates (the exception being R2 of 3D7 + PfAlba3-Ty1), while the timing of the trophozoite transcriptome mapped to 28–30 hpi for the controls and 3D7 + PfAlba4-Ty1 transfectants but to 38–40 hpi for the PfAlba2-Ty1 and PfAlba3-Ty1 transfectants (Fig. 3D). Because the trophozoite stage samples were harvested in the same replication cycle as their matched ring replicate, at 28–30 hpi, and because the Giemsa-stained morphology of all control and transfectant trophozoite samples was similar (Fig. S4), the age of the 3D7 + PfAlba2-Ty1 and 3D7 + PfAlba3-Ty1 trophozoite transcriptome most probably reflects changes in gene expression patterns induced by the ectopic expression of PfAlba2-Ty1 or PfAlba3-Ty1.

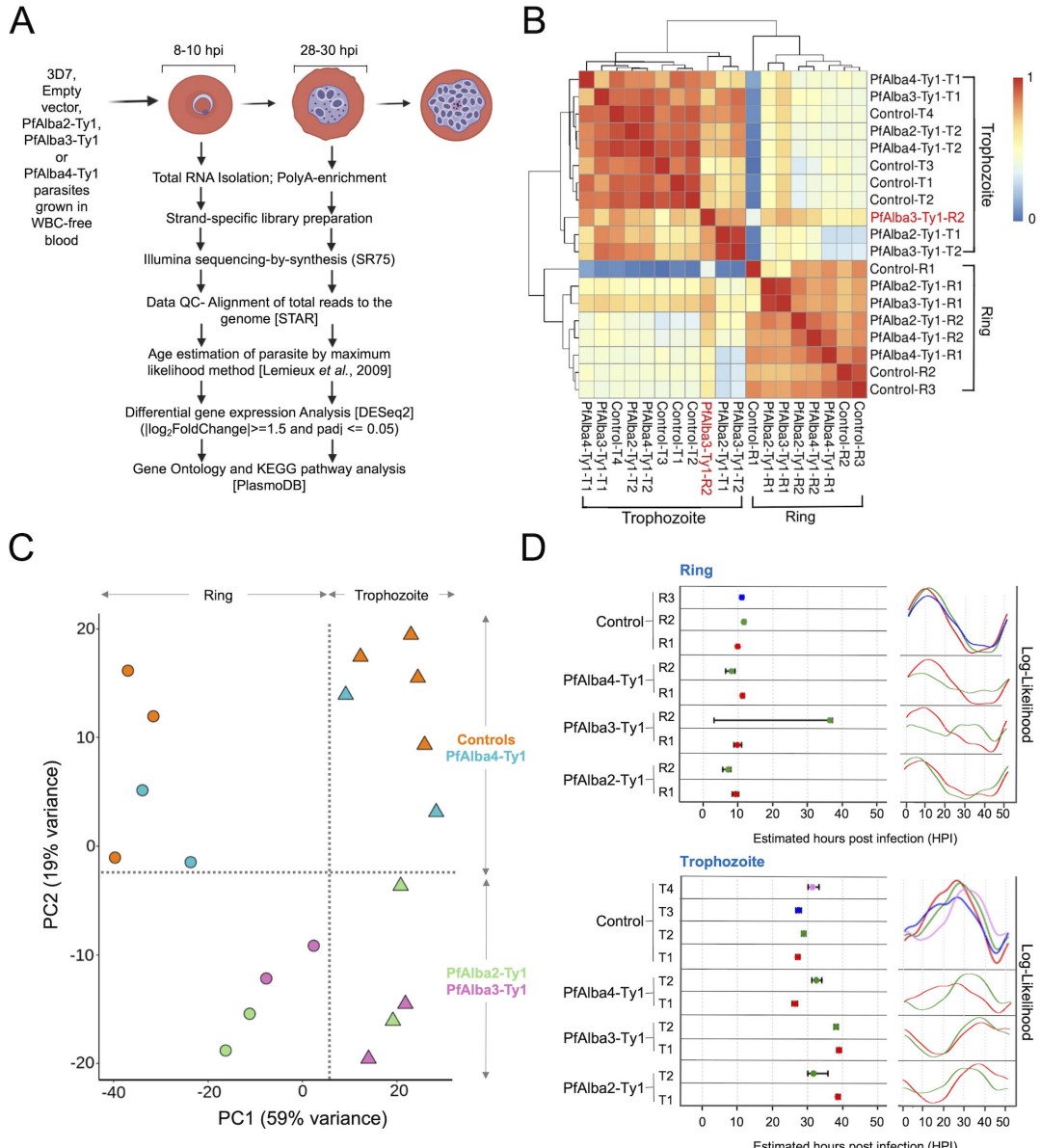

**FIG 3** Transcriptomic data quality assessment by sample clustering demarcates the 3D7 + PfAlba2-ty1 and 3D7 + PfAlba3-Ty1 data sets from controls. (A) Schematic representation of the transcriptomics experiment. 3D7 + empty vector, 3D7 + PfAlba2-Ty1, 3D7 + PfAlba3-Ty1, or 3D7 + PfAlba4-Ty1 parasites were grown in white blood cell (*WBC*)-free blood to ring (8–10 hpi) or trophozoite (28–30 hpi) stages in the presence of 5 μg/mL of blasticidin-S, total RNA harvested, and mRNA enriched and analyzed by strand-specific RNA-seq. Differential gene expression in the Alba-overexpressing lines relative to 3D7 and empty vector transfectants was quantified by DESeq2 analysis. (B) Principal component analysis of normalized read counts of 3D7, 3D7 + empty vector, 3D7 + PfAlba2-Ty1, 3D7 + PfAlba3-Ty1, and 3D7 + PfAlba4-Ty1 ring and trophozoite stage trancriptomes. Meaningful clustering of samples is indicated by dashed lines. (C) Pearson correlation coefficient R analysis of the normalized read counts from RNA-seq analysis of ring (R) and trophozoite (T) stage samples of the 3D7, 3D7 + empty vector, 3D7 + PfAlba2-Ty1, 3D7 + PfAlba3-Ty1, and 3D7 + PfAlba4-Ty1 strains. The color scale indicates the value of the R from 0 to 1. (D) Hours post-infection estimates for the transcriptomic data were obtained by passing the normalized RNA-seq data through the maximum likelihood algorithm developed by Lemieux *et al*. (53). The *left panel* shows the HPI estimates within 95% confidence intervals, while the *right panel* displays the actual likelihoods determined for each sample over the 48-h IED cycle.

## Ectopic PfAlba3-Ty1 expression strongly perturbs the blood stage transcriptome

To identify the changes induced in the *P. falciparum* 3D7 transcriptome upon PfAlba ectopic expression, we performed differential gene expression (DEX) analysis using DESeq2. Based on a |log$_2$(Fold Change)| (*i.e.,* log$_2$FC) cut-off of 1.5 and false discovery

rate-corrected p-value (*i.e.,* padj) of <= 0.05, we shortlisted transcripts that were differentially expressed in the 3D7 + PfAlba2-Ty1, 3D7 + PfAlba3-Ty1, and 3D7 + PfAlba4-Ty1 transcriptomic samples relative to the controls at the ring and trophozoite stages (Fig. 4A). Of the three PfAlbas, PfAlba3 ectopic expression caused the largest changes to the steady-state transcriptome, with 1,521 (682 upregulated and 839 downregulated; approximately 30% of the *P. falciparum* 3D7 transcriptome; Table S5) and 1,047 (587 upregulated and 462 downregulated; approximately 20% of the *P. falciparum* 3D7 transcriptome; Table S6) transcripts being differentially expressed in the ring and trophozoite stages, respectively. This was followed by PfAlba2, which perturbed the steady-state levels of 685 (351 upregulated and 334 downregulated; Table S3) and 368 (187 upregulated and 181 downregulated; Table S4) transcripts during the ring and trophozoite stages, respectively. In contrast, overexpression of PfAlba4 during both stages showed minor perturbations to the transcriptome, with 60 (29 upregulated and 31 downregulated; Table S7) and three transcripts (Table S8) being differentially regulated during the ring and trophozoite stages, respectively, with the former primarily comprising transcripts encoded by multigene families such as *var* and *rifin*.

We next analyzed the DEX overlap between the different transfectants and found that several transcripts were common between the 3D7 + PfAlba2-Ty1 and 3D7 + PfAlba3-Ty1 strains (495 and 351, respectively, during the ring and trophozoite stages); indeed, it appeared that the changes in 3D7 + PfAlba2-Ty1 were, largely, a subset of the changes in 3D7 + PfAlba3-Ty1. This was especially true for the trophozoite samples of 3D7 + PfAlba2-Ty1, where over 95% of the misregulated genes showed the same directionality of change as in the 3D7 + PfAlba3-Ty1 strain, and only 17 genes were unique to 3D7 + PfAlba2-Ty1. This suggested that PfAlba2 and PfAlba3 may function together to regulate a set of transcripts, with PfAlba3 carrying out additional PfAlba2-independent functions. Among all three strains, some overlap in the misregulated transcripts was observed for the ring samples; however, these transcripts turned out to be members of multigene virulence families such as *var*. Upon further comparison of our DEX analysis data to the published DEX list for 3D7 + PfAlba1-Ty1 trophozoite stage samples (38), we observed that, while more than 60% of the DEX genes were commonly misregulated between the 3D7 + PfAlba1-Ty1 and 3D7 + PfAlba3-Ty1 strains, over 300 genes were uniquely misexpressed in each strain (Fig. S6), indicating that PfAlba1 and PfAlba3 perform nonredundant functions during *P. falciparum* blood stage growth.

Lastly, we assessed the overlap between the differentially regulated transcripts in the ring and trophozoite stages of 3D7 + PfAlba2-Ty1 and 3D7 + PfAlba3-Ty1. In keeping with the cyclic expression of gene expression that has been described for *P. falciparum* 3D7 transcripts during IED (3, 6), we observed less than 10% overlap between the two stages for either strain (Fig. 4C). Overall, the DEX analyses of PfAlba2, PfAlba3, and PfAlba4 ectopic expression revealed that PfAlba3, which is 12 kDa in size, contains only the Alba domain and is the smallest among the four previously characterized PfAlbas (2), significantly impacts the *P. falciparum* intra-erythrocytic transcriptome.

## Gene Ontology analysis reveals essential pathways that are up- or downregulated in PfAlba3-overexpressing parasites

Because PfAlba3 overexpression drastically changes the transcriptome of parasites in blood stages, we analyzed the key pathways affected by performing gene set enrichment analyses. First, we used volcano plots to visualize the transcriptome data, which depicts the amount of change (*i.e.,* $\log_2(FC)$) on the x-axis and statistical confidence (*i.e.,* $\log_{10}(p\text{-adj})$) on the y-axis (Fig. 5A and B). Next, for genes with a $|\log_2(FC)|$ cut-off of 1.5 and p-adj <= 0.05, we performed gene ontology (GO) enrichment analysis after removal of genes belonging to multigene families such as *var* and *rifin*. Accordingly, in ring stages, genes that were upregulated corresponded to GO terms such as "peptidase activity," "DNA replication," and "electron transfer activity" (Fig. 5C; *top panel*), while downregulated genes were associated with essential biological processes like "regulation of transcription," "regulation of translational initiation," "ribosome assembly," and "ribosome

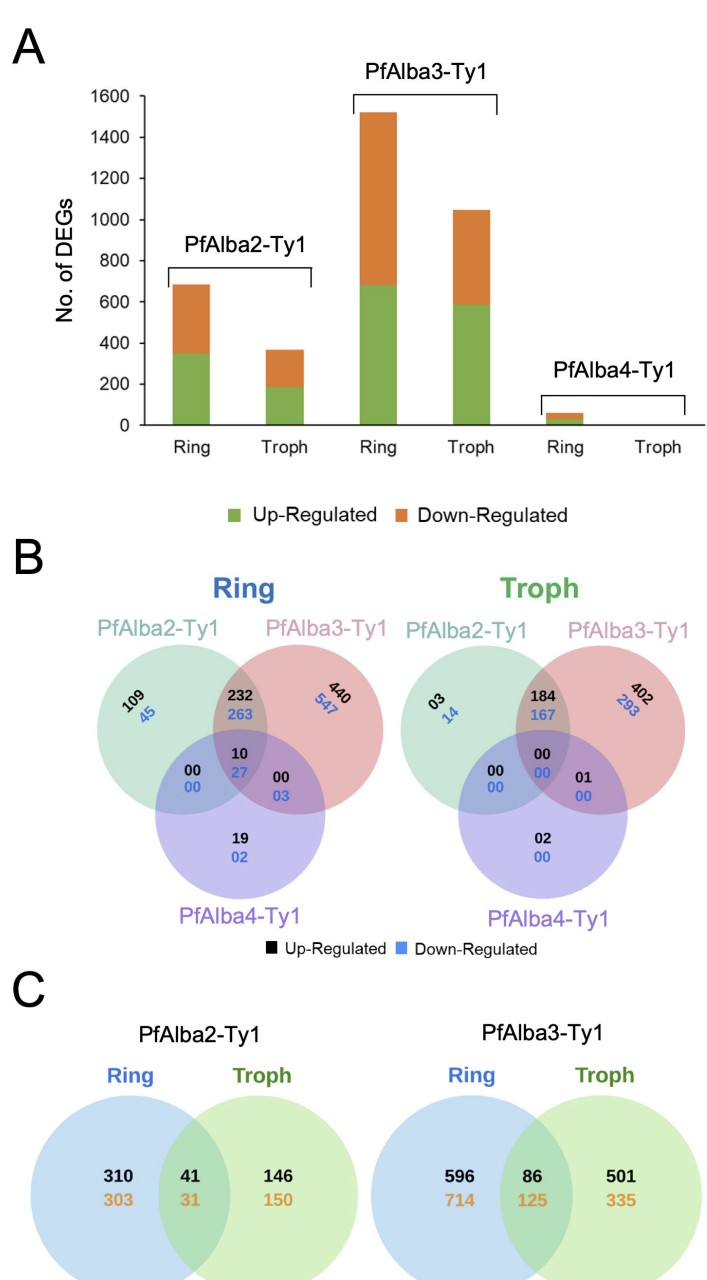

**FIG 4** PfAlba2 and PfAlba3 overexpression causes significant perturbations to the blood stage transcriptome. (A) Bar graph showing the total number of differentially expressed genes (DEGs) that are up- or downregulated in the indicated transgenic *P. falciparum* strain relative to controls. (B) Venn diagrams were used to represent the overlap in DEGs between the various Alba-overexpressing strains during ring and trophozoite stages. (C) Venn diagrams were used to represent the overlap in DEGs between ring and trophozoite stages of either 3D7 + PfAlba2-Ty1 or 3D7 + PfAlba3-Ty1 parasites. Troph = Trophozoite.

biogenesis" (Fig. 5C; *bottom panel*). On the other hand, during the trophozoite stage, upregulated genes encode for rhoptry and cytoskeleton components, which play a role in processes such as "entry into host" and "microtubule-based processes" (Fig. 5D; *top panel*). Strikingly, in this stage, the downregulated genes are associated with "protein folding," "translocation of proteins into host" including the "Maurer's Cleft", and "ribosome biogenesis" (Fig. 5D; *bottom panel*). Taken together, the differential regulation and

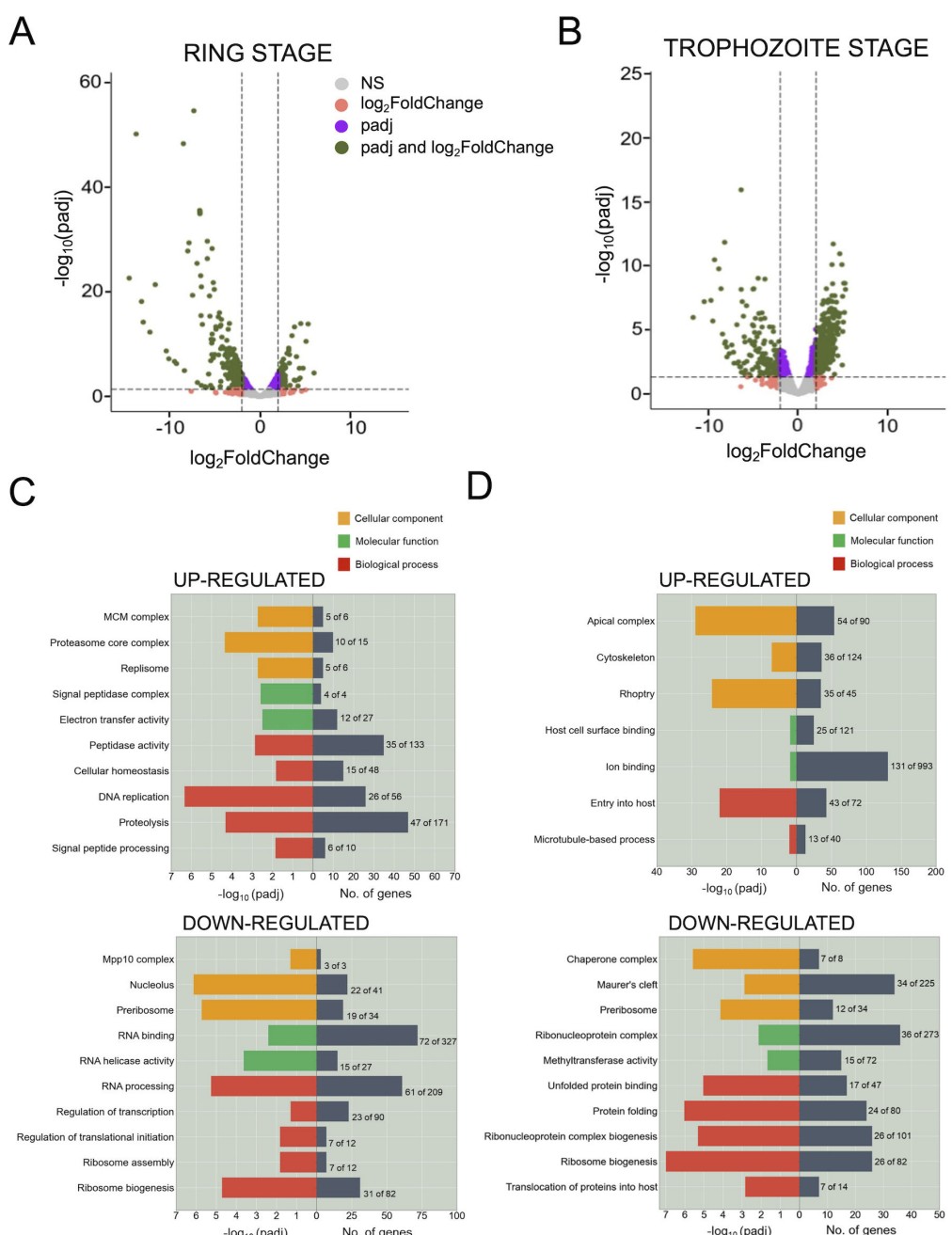

**FIG 5** Mistiming of key cellular developmental processes is linked to PfAlba3 overexpression. (A & B) Volcano plot of differentially expressed genes in (A) ring and (B) trophozoite stages of 3D7 + PfAlba3-Ty1 as compared to controls. Genes with $|log_2FC| > 1.5$ and FDR < 0.05 values were considered to be differentially expressed (green dots). Pink dots indicate genes with $|log_2FC| > 1.5$ but FDR > 0.05, purple dots indicate genes with $|log_2FC| < 1.5$ but FDR < 0.05, and gray dots denote genes that are not changed significantly upon PfAlba3 overexpression. FC = fold change; FDR = false discovery rate. (C & D) Gene Ontology (GO) enrichment analysis in three categories, Cellular Component, Molecular Function, and Biological Process, for significantly up and downregulated genes in the (C) ring and (D) trophozoite stages of 3D7 + PfAlba3-Ty1 parasites. GO enrichment was performed after removing genes belonging to multigene families such as *var* and *rifin*. The number of genes enriched for each GO term relative to background is indicated on the right side of the *y-axis*, while the -$log_{10}$(padj) of each GO term is represented on the left side of the *y-axis*. Note that p-adj is the same as FDR.

mis-timing of several essential pathways, for example, DNA replication being turned on in the ring stage and protein export being disrupted in the trophozoite stage, could

explain the severe growth defects as well as cell cycle misregulation observed for PfAlba3-overexpressing parasites in Fig. 2.

Additionally, we performed a weighted gene co-expression network analysis (WGCNA) of the ring and trophozoite stage transcriptome of the various transfectants and controls (Fig. S7) to identify "hub" genes that may be the first regulatory targets of the PfAlbas (Table S6). From this analysis, we identified MFR2 (major facilitator super-family-related transporter, putative), SEL3 (selenoprotein 3), ACBP2 (acyl-CoA-binding protein, isoform 2), and GBPH (glycophorin-binding protein homolog), along with a few others, as hub genes in the ring stages of PfAlba3-overexpressing cells. Interestingly, all of these genes are significantly upregulated in our DEX analysis. In contrast, the hub genes in 3D7 + PfAlba3-Ty1 trophozoite stages included several members of the inner membrane pellicle complex, a majority of which remained unchanged at the steady-state transcript levels. A future direction would be to determine whether PfAlba3 directly interacts with these genes at the DNA level and/or with their transcripts at the RNA level.

## Global *var* gene repression is observed in parasites overexpressing PfAlba2 or PfAlba3

Since we first discovered the PfAlba proteins as part of a molecular complex that is associated with the *P. falciparum*-specific sub-telomeric repeat Telomere-Associated Repetitive Elements 6 or TARE6, lastly, we checked if excess levels of PfAlba affect the expression of virulence multigene families that lie within the subtelomeric region, adjacent to TARE6. We began with the *var* genes, which reach peak transcript levels in ring stages (8–16 hpi). Upon comparing the Reads Per Kilobase Million (RPKM) values of all 60 *var* genes in the RNA-seq data of the PfAlba transfectants to the empty vector control, we observed a strong repression of all *var* genes in the PfAlba2 or PfAlba3 overexpressing lines, which contrasted with the PfAlba4-overexpressing strain, where

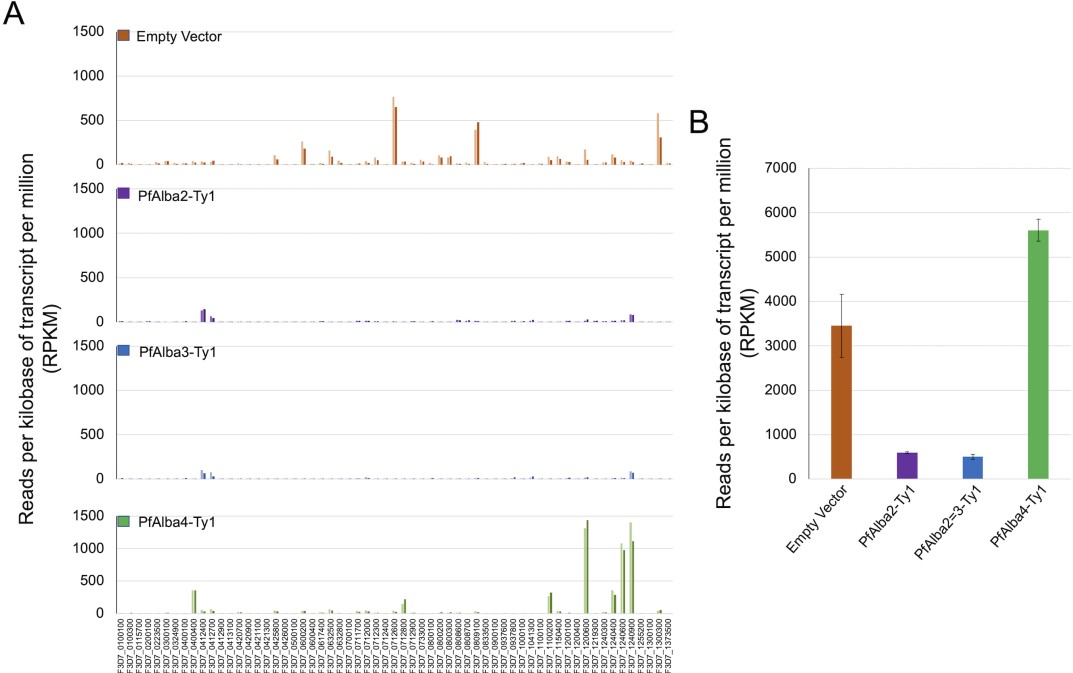

**FIG 6** Global *var* transcriptional repression is apparent in the PfAlba2- and PfAlba3-overexpressing parasites. (A) *var* gene transcriptional profile in ring stages was assessed by RNA-seq of *P. falciparum* 3D7 transfected with either PfAlba2-Ty1, PfAlba3-Ty1, or PfAlba4-Ty1 expression plasmids. Empty vector (pLN-Ty1) transfectants served as a control. The steady-state mRNA levels of all 60 *var* genes is indicated in reads per kilobase of transcript per million or RPKM (*y-axis*). Two replicates for each sample were analyzed. (B) Total mRNA levels (in RPKM) of the *var* gene family was calculated for two ring-stage RNA-seq replicates of each indicated strain. Data represent the mean + STDEV.

three major *var* genes were expressed at high levels (Fig. 6A). Interestingly, 3D7 + PfAlba4-Ty1 showed an overall increase in the total levels of *var* transcripts as compared to the empty vector control (Fig. 6B). Correspondingly, we noticed a higher level of total RUF6 expression in the PfAlba4-overexpressing line (Fig. S8). However, this latter observation needs to be further validated since the polyA-enrichment protocol used for RNA-seq library preparation here captures lower levels of GC-rich RNAs (54).

We also analyzed the expression levels of a second sub-telomeric multigene family *rifin*, which contains ~180 members exhibiting clonal variation and reaches peak transcript levels in trophozoite stages (18–23 hpi). Contrary to the *var* results, significant conclusions could not be made about changes in *rifin* expression between the PfAlba transfectants and the empty vector control (Fig. S9). Taken together, our data point to a strong impact of PfAlba protein levels on *var* gene expression.

## DISCUSSION

Alba proteins are well-conserved and found in all domains of life, including bacteria, archaea, protists, fungi, plants, and animals (56 ). While Alba family members have been extensively investigated as chromatin proteins in archaea (57, 30, 58, 59), they also have the capability to bind to RNA, frequently interacting with double-stranded RNA structures and regulating RNA stability (19, 21, 23, 34). Indeed, in parasitic protozoa, studies on Alba proteins have concentrated on their RNA-binding capacity and not so much on their gene regulatory ability. To address this, the primary objective of this study was to explore the role of three *P. falciparum* Alba domain proteins, PfAlba2, PfAlba3, and PfAlba4, in gene regulation. Given that these proteins are essential and modifications to their genomic locus have met with poor success, ectopic expression of an epitope-tagged version of each PfAlba from a strong heterologous promoter, $P_{cam}$, was used to perturb intracellular protein levels, and the resulting consequences on parasite growth and RNA metabolism were determined.

The first striking observation we made was that the overexpression of PfAlba2-Ty1 and PfAlba3-Ty1 significantly repressed parasite asexual stage growth and interfered with cell cycle progression from the ring to trophozoite to schizont stage. This was not the case with PfAlba4-Ty1, where we observed that excess levels (up to threefold increase) were well-tolerated by the parasite, and unexpectedly, improved parasite IED relative to the empty vector control. While we cannot rule out that the variability in growth phenotypes among the three overexpressor strains could be due to the altered cellular localization of the Ty1-tagged proteins and differences in the relative expression levels of the Ty1-tagged and endogenous PfAlba protein, we are inclined to surmise that the variability stems from the functional diversification of PfAlba2 and PfAlba3 relative to PfAlba4. In fact, the overexpression of PfAlba2 and PfAlba3 in the context of their respective wild-type genomic copy appears to exert a dominant negative effect, which is not the case with PfAlba4. A second explanation for the growth defect of 3D7 + PfAlba3-Ty1 arises from a recent report that demonstrated the *in vitro* endonuclease activity of recombinant PfAlba3 in the presence of divalent metal ions such as $Ca^{2+}$ and $Mg^{2+}$ at abasic sites in double-stranded DNA (41). Therefore, when this protein is overexpressed, it may damage DNA by creating double-stranded breaks, in turn causing genomic instability and parasite death. Whether PfAlba2 also possesses endonuclease activity remains an open question.

Second, of the three PfAlbas, sub-lethal overexpression of PfAlba3—which contains only a ~ 12-kDa Alba domain—showed the strongest perturbation of the steady-state transcriptomes of both ring and trophozoite stages. While PfAlba2 overexpression also resulted in transcriptional dysregulation, it was to a lesser extent, and a majority of the deregulated transcripts in 3D7 + PfAlba2-Ty1 were a subset of those impacted by PfAlba3. In contrast, PfAlba4 overexpression had a negligible impact on the parasite steady-state transcriptome. This result initially surprised us because phylogenetic studies have shown that the Alba domains of PfAlba1 and PfAlba2 are more similar, with PfAlba3 grouping within the same clade as PfAlba4's Alba domain (15, 17, 19); our recent analyses

have also supported this grouping (56). However, given the dramatically different transcriptional phenotypes of the 3D7 + PfAlba3-Ty1 and 3D7 + PfAlba4-Ty1 strains, we can conclude that primary sequence similarity alone does not determine Alba domain function. For example, the overlap of PfAlba2 and PfAlba3 targets could arise from their functioning as a heterodimer during asexual stages, while in the case of PfAlba4, its functions may be determined by its C-terminal MAF1 domain (discussed below), its potential to homodimerize, or through its heterodimerization with PfAlba1, PfAlba5, or PfAlba6. In fact, the ability of Alba domain proteins to homo- or hetero-dimerize has been previously reported in archaea and plants and is strongly implicated in chromatin compaction in archaea (26, 28, 59, 60). Moreover, in the mouse malaria parasite *P. yoelli*, PyAlba4 was shown to regulate sexual stage development by interacting with three other Alba proteins, PyAlba1, PyAlba2, and PyAlba3 (61).

Third, in ring stages of 3D7 + PfAlba3-Ty1, upregulated transcripts included 26 genes related to DNA replication, such as the DNA helicase subunits MCM3/4/5/7, RPA1 (replication protein A1), DNA polymerase delta, and RFC4 (replication factor C subunit 4) and 47 genes that participate in cellular proteolysis including several proteasomal subunits. These genes are typically expressed at the onset of replication in the trophozoite stage (~30 hpi; https://plasmodb.org). Therefore, their early expression in ring stages of the 3D7 + PfAlba3-Ty1 strain could partially explain the observed developmental defect. Similarly, in the trophozoite stage, transcripts of 43 invasion-related genes such as MSP2 (merozoite surface protein 2), RON6 (rhoptry neck protein 6), RhopH3 (rhoptry-associated protein 3), and AMA1 (apical membrane antigen 1) were upregulated. These genes are typically turned on toward the start of the schizont stage (~40 hpi; https://plasmodb.org) by a specialized transcription factor of the ApiAP2 family, AP2-I (62). Interestingly, AP2-I levels are strongly upregulated in the ring stage of 3D7 + PfAlba3-Ty1, but not in the trophozoite stage. This suggests that the premature expression of invasion genes could be an indirect effect of higher PfAlba3 in the ring stages. This also suggests that PfAlba proteins may regulate the levels of key transcription factors of the ApiAP2 family, not just in the asexual stages but also during other stages of the developmental cycle. Nonetheless, we cannot rule out that PfAlba3 directly binds to and stabilizes the transcripts that show increased steady-state levels. Such an observation has been made in *Leishmania* where LiAlba20 binds to the 3'UTR of the mRNA of amastin, a virulence factor, and stabilizes it in amastigote stages (63, 24).

In general, a decrease in steady-state transcript levels of a gene could be either due to a decrease in transcription or a shift in splicing patterns or through a reduction in mRNA stability; proteins that bind to DNA and RNA such as the PfAlbas could impact all of these processes directly as well as indirectly. In this context, the genes which were downregulated in 3D7 + PfAlba3-Ty1 cells were particularly interesting. In ring stages, a majority of the downregulated genes were linked to ribosome biogenesis and RNA regulation, including translation initiation, while in the trophozoite stages, they were linked to protein folding and homeostasis, in addition to ribosome biogenesis. These results strongly implicate PfAlba3 as a negative regulator of ribosome levels, either through transcriptional repression of the genes encoding ribosomal components or through mRNA destabilization. Consequently, in the presence of excess PfAlba3, *P. falciparum* may face challenges in synthesizing sufficient amounts of proteins necessary for essential developmental processes, growth, and survival. In fact, Alba domain proteins have been linked to translation regulation in other organisms like *T. brucei, L. infantum,* and *T. gondii* (21, 23, 24). Moreover, many of the Alba-domain proteins in *P. falciparum* are polysome-associated (13), and, in female gametocytes of the mouse malaria parasite *P. berghei*, PbAlba1, PbAlba2, and PbAlba3 co-purified with the DOZI (Dead-box RNA helicase) and CITH (homolog of worm CAR-I and fly Trailer Hitch) complex, which is required for translational repression of over 300 maternal mRNAs (16).

Finally, apart from the transcriptional dysregulation of major cellular processes, PfAlba2 and PfAlba3 overexpression in ring stages globally downregulated *var* transcript levels. The ~60-member *var* multigene family, many of which are located at

sub-telomeric regions of chromosomes, undergoes mutually exclusive expression, a phenomenon that is controlled by a complex interplay of epigenetic factors and ncRNAs (36, 37, 64–68). Over the years, several lines of evidence have connected the PfAlbas to this process. We first identified the PfAlbas as part of a protein complex that binds to repeat sequences found within sub-telomeric regions, proximal to *var* genes (17). Additionally, PfAlba3 is present at *var* perinuclear foci (69), while PfAlba1, PfAlba2, and PfAlba3 are known to associate with regulatory sequences within *var* introns (36, 37). PfAlba3 has also been shown to interact with epigenetic regulators of *var* expression such as PfSET10 (70) and PfSir2A (69). Just in the past year, PfAlba2 was shown to bind to the RUF6 ncRNA, which is an activator of *var* expression (40). Interestingly, we observed that the steady-state mRNA levels of PfHP1, which is a key regulator of *var* silencing through its interaction with the histone H3 lysine 9 trimethyl mark (71, 72), is upregulated in the ring stages of 3D7 + PfAlba3-Ty1 (Table S5). Therefore, the *var* repression observed in this strain may be due to elevated PfHP1 levels. On the other hand, PfHP1 levels are not upregulated in the 3D7 + PfAlba2-Ty1 strain, neither did we observe changes in the levels of other known epigenetic regulators (Table S3). Nonetheless, it is possible that *var* repression may be a secondary effect of cell cycle dysregulation in the 3D7 + PfAlba2-Ty1 and 3D7 + PfAlba3-Ty1 strains, necessitating a detailed exploration of Alba function in the context of *var* gene regulation.

Overall, our findings with regard to PfAlba2 and PfAlba3 overexpression phenocopied our previous results for PfAlba1 (38), whereas PfAlba4 overexpression manifested differently: Ty1-tagged PfAlba4 localized primarily to the cytoplasm, and its overexpression promoted parasite asexual growth, did not induce significant IED transcriptomic changes, and appeared to increase *var* mRNA levels. This is interesting because PfAlba4, at 42 kDa, is the biggest of the six PfAlbas, and in addition to the N-terminal Alba domain contains, at its C-terminus, a 30-kDa MAF1_Alba4_C domain (where MAF1 is the C-terminal domain of mitochondrial association factor 1 from *T. gondii*). Although the exact function of this domain is not known, our recent FoldSeek (73) analysis revealed it to be similar to histone macro H2A.1 from *Mus musculus, Homo sapiens,* and other vertebrates. This implies that the function of PfAlba4 in asexual stages may be diversified by its C-terminal end. For instance, a recent study identified PfAlba4 as a core member of the cytoplasmic nonsense-mediated decay (NMD) complex, where it directly interacts with the nonessential proteins PfUpf1 and PfUpf2 (74). Given that NMD does not contribute significantly to *P. falciparum* gene regulation (74), it is not surprising that PfAlba4 overexpression is not disruptive to the parasite IED transcriptome. However, it remains to be seen if the interaction of PfAlba4 with PfUpf1 and PfUpf2 is mediated by the MAF1 domain. Additionally, PfAlba4 complexes with epigenetic factors such as PfBDP1 (75) and other proteins (76) during the IED, which could dictate its functionality.

To conclude, this study supports the view that the DNA-/RNA-binding Alba family is a master regulator of *P. falciparum* gene expression. This warrants a comprehensive evaluation of PfAlbas' interactions with DNA and RNA targets and the outcomes of these interactions. Particular areas of focus could include developmental stages beyond the IED, the effect of nucleobase modifications on Alba-nucleic acid binding, and the hitherto uncharacterized PfAlba5 and PfAlba6 proteins. Additionally, the recent findings in *Arabidopsis* that AtALBA1 and AtALBA2 bind to DNA:RNA-hybrid structures called R-loops as a heterodimer/heteropolymer and shield genomic DNA from damage and instability (26) could open up a new line of investigation for the PfAlbas as well. Ultimately, given their essentiality and sequence divergence from human Alba domain-containing proteins (56), the PfAlba protein family could be developed as a potential target for antimalarial design and discovery.

## ACKNOWLEDGMENTS

The authors acknowledge Illumina sequencing support from Odile Sismeiro and Jean-Yves Coppee of the Institut Pasteur NGS Core. This work was supported by the Ramalingaswami Re-entry Fellowship (BT/RLF/Re-entry36/2017) awarded by the

Department of Biotechnology, India, and the European Research Council's Marie Sklodowska-Curie International Incoming Fellowship (FP7-MC-IIF-302451) to S.S.V., and a European Research Council Advanced Grant (PlasmoEscape 250320) and the French Parasitology consortium ParaFrap (ANR-11-LABX0024) to A.S. D.A. acknowledges PhD support from the Department of Biotechnology, India, under the DBT-JRF scheme. Work in the S.S.V. lab is supported by the Department of IT, BT, S&T of the Government of Karnataka, India.

## AUTHOR AFFILIATIONS

[1]Manipal Academy of Higher Education, Manipal, Karnataka, India
[2]Institute of Bioinformatics and Applied Biotechnology, Bengaluru, Karnataka, India
[3]Unité de Biologie des Interactions Hôte-Parasite, Institut Pasteur, Paris, France
[4]CNRS ERM9195, Paris, France
[5]INSERM U1201, Paris, France

## AUTHOR ORCIDs

Dimple Acharya http://orcid.org/0009-0002-7863-1868
Shruthi Sridhar Vembar http://orcid.org/0000-0002-7497-6378

## AUTHOR CONTRIBUTIONS

Dimple Acharya, Formal analysis, Investigation, Methodology, Validation, Visualization, Writing – original draft, Writing – review and editing, Data curation | Anitha Nagaraj Bavikatte, Formal analysis, Investigation, Methodology, Writing – review and editing | Vishnu Vinayak Ashok, Formal analysis, Investigation, Methodology, Writing – review and editing | Shubhada R. Hegde, Formal analysis, Investigation, Writing – review and editing | Cameron Ross Macpherson, Formal analysis, Investigation, Methodology, Writing – review and editing | Artur Scherf, Funding acquisition, Project administration, Writing – review and editing | Shruthi Sridhar Vembar, Conceptualization, Formal analysis, Funding acquisition, Investigation, Methodology, Project administration, Supervision, Validation, Visualization, Writing – original draft, Writing – review and editing

## DATA AVAILABILITY

The data sets, *i.e.,* fastq files, supporting the results of this article are available in the NCBI Short Read Archive [BioProject ID PRJNA1089486].

## ADDITIONAL FILES

The following material is available online.

### Supplemental Material

**Supplemental material (Spectrum00885-24-s0001.pdf).** Table S1; Figures S1 to S9
**Table S2 (Spectrum00885-24-s0002.xlsx).** Consolidated HTSeq-generated read counts for all *P. falciparum* genes in the transcriptomic datasets generated in this study.
**Table S3 (Spectrum00885-24-s0003.xlsx).** List of differentially expressed genes in Ring stages of the 3D7+PfAlba2-Ty1 strain.
**Table S4 (Spectrum00885-24-s0004.xlsx).** List of differentially expressed genes in Trophozoite stages of the 3D7+PfAlba2-Ty1 strain.
**Table S5 (Spectrum00885-24-s0005.xlsx).** List of differentially expressed genes in Ring stages of the 3D7+PfAlba3-Ty1 strain.
**Table S6 (Spectrum00885-24-s0006.xlsx).** List of differentially expressed genes in Trophozoite stages of the 3D7+PfAlba3-Ty1 strain.
**Table S7 (Spectrum00885-24-s0007.xlsx).** List of differentially expressed genes in Ring stages of the 3D7+PfAlba4-Ty1 strain.

**Table S8 (Spectrum00885-24-s0008.xlsx).** List of differentially expressed genes (significantly upregulated) in Trophozoite stages of the 3D7+PfAlba4-Ty1 strain.

Open Peer Review

**PEER REVIEW HISTORY (review-history.pdf).** An accounting of the reviewer comments and feedback.

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
