## [Reviewer comments · Microbiology Spectrum]

Microbiology Spectrum

Ectopic overexpression of *Plasmodium falciparum* DNA/RNA-binding Alba proteins misregulates virulence gene homeostasis during asexual blood development

Dimple Acharya, Anitha Bavikatte, Vishnu Ashok, Shubhada Hegde, Cameron Macpherson, Artur Scherf, and Shruthi Vembar

Corresponding Author(s): Shruthi Vembar, Institute of Bioinformatics and Applied Biotechnology

Review Timeline:

Submission Date:	April 6, 2024
Editorial Decision:	April 22, 2024
Revision Received:	October 29, 2024
Accepted:	November 27, 2024

Editor: Björn Kafsack

Reviewer(s): Disclosure of reviewer identity is with reference to reviewer comments included in decision letter(s). The following individuals involved in review of your submission have agreed to reveal their identity: Francesca Florini (Reviewer #1)

Transaction Report:

DOI: <https://doi.org/10.1128/spectrum.00885-24>

Re: Spectrum00885-24 (Ectopic overexpression of *Plasmodium falciparum* DNA/RNA-binding Alba proteins misregulates virulence gene homeostasis during asexual blood development)

Dear Dr. Shruthi Sridhar Vembar:

Thank you for the privilege of reviewing your work. Below you will find my comments, instructions from the Spectrum editorial office, and the reviewer comments.

Revision Guidelines

Sincerely,
Björn Kafsack
Editor
Microbiology Spectrum

Reviewer #1 (Comments for the Author):

In this study, Acharya and colleagues overexpress three members of the Alba protein family-namely, PfAlba2, PfAlba3, and PfAlba4-and determine their effects on parasite growth and the transcriptome. They demonstrate that overexpression of PfAlba2 and PfAlba3, but not PfAlba4, significantly impairs parasite growth and causes widespread perturbations in mRNA levels of multiple genes during the erythrocytic cycle.

The manuscript is generally well-written and clear. I have a few comments and questions for the authors to address:

Figure 1C/D: Given their previous publication, the authors have access to antibodies for at least PfAlba2 and PfAlba4. It would be beneficial to add Western blots and immunofluorescence assays (IFAs) using these antibodies to compare the steady-state levels of Albas with their levels upon overexpression, as well as to compare localization in control and overexpression cell lines. For instance, in a previous publication by the authors, PfAlba2 was localized to the nuclear periphery at the ring stage, whereas the Ty-tagged version appears to localize throughout the entire nucleus. Could the authors speculate on this difference? Moreover, the overexpression of PfAlba3 appears to be much more pronounced than that of the others. Could this be why the effect of Alba3 overexpression is stronger?

Figure 1C: Was this Western blot performed in the presence of Blasticidin? It would be informative to see a comparison between 0 and 5 µg/ml of Blasticidin for PfAlba2 and PfAlba3 (if they were unable to collect samples at higher concentrations), especially since the rest of the transcriptomics work was conducted at 5 µg/ml of Blasticidin. The same applies to the IFA.

Figure 2A: The authors should clarify in the figure legend what the red arrows indicate.

Figure 2B: Did the authors perform Giemsa stains of these different time points? What is the morphology of these parasites?

There are a couple of ambiguities regarding the RNA-Seq data: One of the two replicates of PfAlba3-Ty at the ring stage is markedly different from the other. Was this sample excluded from subsequent analyses?

In the Materials and Methods section for RNA-Seq, the authors mention using two controls: untransfected 3D7 and 3D7+empty vector, analyzing three and four biological replicates, respectively (totaling seven). However, in Figure 3B only four controls are shown, and in Figures 3C/D, there are three controls at the ring stage and four at the trophozoite stage, with no explanation provided for discarding some.

Figure 6: How can the authors be certain that the downregulation of var genes is not a secondary effect of cell cycle dysregulation?

Lines 437-438: The authors state there is no effect on rifins, but it appears to me that rifins are equally downregulated as vars in PfAlba2 and PfAlba3 overexpression (Supplementary Figure 6).

Reviewer #2 (Comments for the Author):

Acharya et al report a very interesting observation demonstrating ectopic overexpression of Alba domain proteins in *P. falciparum* mis-regulates virulence gene homeostasis during asexual blood development. Prior to acceptance for publication, the authors should endeavor to address the following concerns:

Comments:

1. In line no. 161: Should it be pPfAlba4-Ty1C or pPfAlba3-Ty1C?
2. Provide histogram plot of the Flow cytometry data in the supplementary
3. In Fig 1 C: Show molecular weight markers.
4. Fig 1 D: Show DIC and merged panel for all the stages (DAPI and Ty1) along with 3D sectioning mentioning the Pearson coefficient for nuclear localization stages. The subcellular localization of the protein within the nucleus and cytoplasm cannot be established by confocal data alone, need further experimentation to support the claim.
5. Alba domain proteins have been reported to be modulated by different PTMs (Acetylation, methylation, Phosphorylation). So, upon overexpression is there any indication of differential status of expression of family of genes (broadly) modulating PTM in Alba domain proteins?
6. In line 114: The functional validation of PfAlba6 has been reported (Nag et al.,2024) that is not mentioned.

MICROBIOLOGY SPECTRUM – Acharya et al.

In this study, Acharya and colleagues overexpress three members of the Alba protein family—namely, PfAlba2, PfAlba3, and PfAlba4—and determine their effects on parasite growth and the transcriptome. They demonstrate that overexpression of PfAlba2 and PfAlba3, but not PfAlba4, significantly impairs parasite growth and causes widespread perturbations in mRNA levels of multiple genes during the erythrocytic cycle.

The manuscript is generally well-written and clear. I have a few comments and questions for the authors to address:

Figure 1C/D: Given their previous publication, the authors have access to antibodies for at least PfAlba2 and PfAlba4. It would be beneficial to add Western blots and immunofluorescence assays (IFAs) using these antibodies to compare the steady-state levels of Albas with their levels upon overexpression, as well as to compare localization in control and overexpression cell lines. For instance, in a previous publication by the authors, PfAlba2 was localized to the nuclear periphery at the ring stage, whereas the Ty-tagged version appears to localize throughout the entire nucleus. Could the authors speculate on this difference?

Moreover, the overexpression of PfAlba3 appears to be much more pronounced than that of the others. Could this be why the effect of Alba3 overexpression is stronger?

Figure 1C: Was this Western blot performed in the presence of Blasticidin? It would be informative to see a comparison between 0 and 5 µg/ml of Blasticidin for PfAlba2 and PfAlba3 (if they were unable to collect samples at higher concentrations), especially since the rest of the transcriptomics work was conducted at 5 µg/ml of Blasticidin. The same applies to the IFA.

Figure 2A: The authors should clarify in the figure legend what the red arrows indicate.

Figure 2B: Did the authors perform Giemsa stains of these different time points? What is the morphology of these parasites?

There are a couple of ambiguities regarding the RNA-Seq data: One of the two replicates of PfAlba3-Ty at the ring stage is markedly different from the other. Was this sample excluded from subsequent analyses?

In the Materials and Methods section for RNA-Seq, the authors mention using two controls: untransfected 3D7 and 3D7+empty vector, analyzing three and four biological replicates, respectively (totaling seven). However, in Figure 3B only four controls are shown, and in Figures 3C/D, there are three controls at the ring stage and four at the trophozoite stage, with no explanation provided for discarding some.

Figure 6: How can the authors be certain that the downregulation of *var* genes is not a secondary effect of cell cycle dysregulation?

Lines 437-438: The authors state there is no effect on *rifins*, but it appears to me that *rifins* are equally downregulated as *vars* in PfAlba2 and PfAlba3 overexpression (Supplementary Figure 6).

Acharya et al report a very interesting observation demonstrating ectopic overexpression of Alba domain proteins in *P. falciparum* mis-regulates virulence gene homeostasis during asexual blood development. Prior to acceptance for publication, the authors should endeavor to address the following concerns:

Comments:

1. In line no. 161: Should it be pPfAlba4-Ty1C or pPfAlba3-Ty1C?
2. Provide histogram plot of the Flow cytometry data in the supplementary
3. In Fig 1 C: Show molecular weight markers.
4. Fig 1 D: Show DIC and merged panel for all the stages (DAPI and Ty1) along with 3D sectioning mentioning the Pearson coefficient for nuclear localization stages. The subcellular localization of the protein within the nucleus and cytoplasm cannot be established by confocal data alone, need further experimentation to support the claim.
5. Alba domain proteins have been reported to be modulated by different PTMs (Acetylation, methylation, Phosphorylation). So, upon overexpression is there any indication of differential status of expression of family of genes (broadly) modulating PTM in Alba domain proteins?
6. In line 114: The functional validation of PfAlba6 has been reported (Nag et al.,2024) that is not mentioned.

25 October 2024

Re: Spectrum00885-24

Dear Prof. Bjorn Kafsack,

Thank you for sending the comments of the reviewers. In the following pages, we have responded to all Reviewer Comments using **blue font colour**. The accompanying changes in the main manuscript are highlighted in blue.

We thank both reviewers for their comments. By addressing them, we believe that the paper is stronger.

We hope that with this revision, you will accept our manuscript for publication in your esteemed journal.

Thank you very much for your patience with this manuscript and we once gain apologise for the delay in resubmission.

Best regards,
Shruthi Vembar

Reviewer #1 (Comments for the Author):

In this study, Acharya and colleagues overexpress three members of the Alba protein family—namely, PfAlba2, PfAlba3, and PfAlba4—and determine their effects on parasite growth and the transcriptome. They demonstrate that overexpression of PfAlba2 and PfAlba3, but not PfAlba4, significantly impairs parasite growth and causes widespread perturbations in mRNA levels of multiple genes during the erythrocytic cycle.

The manuscript is generally well-written and clear. I have a few comments and questions for the authors to address:

1. Figure 1C/D: Given their previous publication, the authors have access to antibodies for at least PfAlba2 and PfAlba4. It would be beneficial to add Western blots and immunofluorescence assays (IFAs) using these antibodies to compare the steady-state levels of Albas with their levels upon overexpression, as well as to compare localization in control and overexpression cell lines. For instance, in a previous publication by the authors, PfAlba2 was localized to the nuclear periphery at the ring stage, whereas the Ty-tagged version appears to localize throughout the entire nucleus. Could the authors speculate on this difference?

Author response:

We have included the requested western blot images of 3D7+empty vector vs 3D7+PfAlba2-Ty1 lysates probed with anti-PfAlba2 antibodies and 3D7+empty vector vs 3D7+PfAlba2-Ty1 lysates probed with anti-PfAlba3 antibodies in Supplementary Figures S2A and S2B, respectively. These results are alluded to on Page 12, Lines 290-291.

Note that we have included the 3D7+PfAlba4-Ty1 western blot images as parts C and D of Supplementary Figure S2; in the original submission, these were parts A and B of Supplementary Figure S1.

For IFAs, we have introduced a new supplementary figure, Supplementary Figure S1. This contains IFA images of 3D7+empty vector and 3D7+PfAlba2-Ty1 strains stained with anti-PfAlba2 antibodies, and 3D7+empty vector and 3D7+PfAlba4-Ty1 strains stained with anti-PfAlba4 antibodies. Although the anti-PfAlba3 antibodies worked in western blotting, they were unsuccessful in IFAs despite our many efforts. Therefore, we cannot comment on the cellular localization of endogenous PfAlba3 in the 3D7+PfAlba3-Ty1 strain. All of these results are cited on Page 11, Lines 265-272 and discussed in the Discussion section on Page 18, Lines 484-488.

2. Moreover, the overexpression of PfAlba3 appears to be much more pronounced than that of the others. Could this be why the effect of Alba3 overexpression is stronger?

Author response:

We agree with the reviewer that in the blot shown in Figure 1C, Alba3-Ty1 appears to be strongly overexpressed. One explanation for this could be its smaller size. Another explanation could be the lower expression levels/poorer detection of PfAlba2-Ty1, *i.e.*, when 3D7+PfAlba2-Ty1 lysates are run alongside the 3D7+PfAlba2-Ty1 lysates on the same gel, the PfAlba3-Ty1 signal is saturated at an exposure where the PfAlba2-Ty1 signal can be detected at sufficient intensity. This is also apparent in Supplementary Figures S2A and S2B.

3. Figure 1C: Was this Western blot performed in the presence of Blasticidin? It would be informative to see a comparison between 0 and 5 µg/ml of Blasticidin for PfAlba2 and PfAlba3 (if they were unable to collect samples at higher concentrations), especially since the rest of the transcriptomics work was conducted at 5 µg/ml of Blasticidin. The same applies to the IFA.

Author response:

Yes, the Figure 1C western blot and Figure 1D IFAs were performed in the presence of 5 µg/ml Blasticidin-S. This is now mentioned in the figure legend.

As requested by the reviewer, we compared the anti-Ty1 and anti-PfAlba2 signals for 3D7+PfAlba2-Ty1 at 0 and 5 µg/ml Blasticidin-S, and the anti-Ty1 and anti-PfAlba3 signals for 3D7+PfAlba3-Ty1 at 0 and 5 µg/ml Blasticidin-S (Supplementary Figures S2A and S2B, respectively). We did not observe a drastic difference in the levels of the tagged protein at the two Blasticidin-S concentrations and hence did not repeat the IFAs at the lower concentration. These results are alluded to on Page 12, Lines 290-291.

4. Figure 2A: The authors should clarify in the figure legend what the red arrows indicate.

Author response:

We have inserted the following statement in the figure legend - “Red arrows indicate growth profiles of the strains at 10 µg/ml BS concentration.”

We have also referred to this in the Results - on Page 12, Lines 285-286.

Figure 2B: Did the authors perform Giemsa stains of these different time points? What is the morphology of these parasites?

Author response:

We have included GIEMSA images of ring (10-14 hours post invasion or hpi), trophozoite (28-32 hpi) and schizont (40-44 hpi) stages of the different strains as a new Supplementary Figure S4 and indicate that there are no apparent changes in morphology on Page 12, Lines 305-306.

There are a couple of ambiguities regarding the RNA-Seq data: One of the two replicates of PfAlba3-Ty at the ring stage is markedly different from the other. Was this sample excluded from subsequent analyses?

Author response:

We agree with the reviewer that one of the two 3D7+PfAlba3-Ty1 ring stage replicates behaves differently in the Pearson Correlation Coefficient (PCC) and time estimate analyses, although it is not an outlier in PCA (Figure 3). In fact, we performed RNA-seq of a total of four ring stage replicates of 3D7+PfAlba3-Ty1 and found that two of the replicates had very low read counts; these had to be discarded (we have included this information in the Materials and Methods section on Page 9, Lines 220-223).

Of the two 3D7+PfAlba3-Ty1 ring replicates that were retained, in PCC, one appeared to be more similar to 3D7+PfAlba2-Ty1 Troph rep 1 and 3D7+PfAlba3-Ty1 Troph rep 2. However, when we looked closely, we found that its PCC values were as follows:

3D7+PfAlba3-Ty1 Ring rep 2 to 3D7+PfAlba2-Ty1 Troph rep 1 = 0.82

3D7+PfAlba3-Ty1 Ring rep 2 to 3D7+PfAlba3-Ty1 Troph rep 2 = 0.82

3D7+PfAlba3-Ty1 Ring rep 2 to 3D7+PfAlba2-Ty1 Ring rep 1 = 0.82

Therefore, this sample is as correlated to 3D7+PfAlba3-Ty1 Ring rep 1 as it is to some of the trophozoite replicates. Overall, based on its behaviour in PCA, we included it for subsequent analyses. We have mentioned this on Page 13, Lines 328-330.

In the Materials and Methods section for RNA-Seq, the authors mention using two controls: untransfected 3D7 and 3D7+empty vector, analyzing three and four biological replicates, respectively (totaling seven). However, in Figure 3B only four controls are shown, and in Figures 3C/D, there are three controls at the ring stage and four at the trophozoite stage, with no explanation provided for discarding some.

Author response:

We thank the reviewer for bringing this up and realise that we have not described the number of replicates accurately in the Materials and Methods. We sequenced two biological replicates each for 3D7 ring, 3D7 trophozoite, 3D7+empty vector ring and 3D7+empty vector trophozoite. Due to low read counts, one of the 3D7 ring replicates had to be excluded from further analysis. We have corrected this information in the Materials and Methods section on Page 9, Lines 220-223. Therefore, in Figures 3B, 3C and 3D, there are three ring controls and four trophozoite controls, totalling to seven.

Figure 6: How can the authors be certain that the downregulation of var genes is not a secondary effect of cell cycle dysregulation?

Author response:

We thank the reviewer for bringing this up. Indeed, we cannot exclude the possibility that *var* repression is a secondary effect of cell cycle dysregulation in the 3D7+PfAlba2-Ty1 and 3D7+PfAlba3-Ty1 strains. We have included this as a discussion point on Page 21, Lines 580-583.

Lines 437-438: The authors state there is no effect on rifins, but it appears to me that rifins are equally downregulated as vars in PfAlba2 and PfAlba3 overexpression (Supplementary Figure 6).

Author response:

While one of the two replicates showed lower rifin levels for 3D7+PfAlba2-Ty1 and 3D7+PfAlba3-Ty1 strains, the other replicate mirrored the profiles of 3D7+empty vector and 3D7+PfAlba4-Ty1. Therefore, we cannot make a strong statement regarding rifin dysregulation. We have retained our original conclusions on Page 17, Lines 457-462.

Overall, we would like to thank the reviewer for their insightful comments. These have truly helped us to improve the quality of the manuscript.

Reviewer #2 (Comments for the Author):

Acharya et al report a very interesting observation demonstrating ectopic overexpression of Alba domain proteins in *P. falciparum* mis-regulates virulence gene homeostasis during asexual blood development. Prior to acceptance for publication, the authors should endeavor to address the following concerns:

Comments:

1. In line no. 161: Should it be pPfAlba4-Ty1C or pPfAlba3-Ty1C?

Author response:

We thank the reviewer for catching this typo. We have edited the text - Page 7, Lines 163.

2. Provide histogram plot of the Flow cytometry data in the supplementary

Author response:

We have included a new Supplementary Figure S3 containing a sample histogram plot corresponding to one of the datapoints of Figure 2B.

3. In Fig 1 C: Show molecular weight markers.

Author response:

We have included molecular weight markers in Figure 1C.

4. Fig 1 D: Show DIC and merged panel for all the stages (DAPI and Ty1) along with 3D sectioning mentioning the Pearson coefficient for nuclear localization stages. The subcellular localization of the protein within the nucleus and cytoplasm cannot be established by confocal data alone, need further experimentation to support the claim.

Author response:

We have edited Figure 1D to include DIC images as a merged panel. We performed 3D sectioning and have included Manders' Coefficient and PCC values in new Supplementary Figure S1B.

5. Alba domain proteins have been reported to be modulated by different PTMs (Acetylation, methylation, Phosphorylation). So, upon overexpression is there any indication of differential status of expression of family of genes (broadly) modulating PTM in Alba domain proteins?

Author response:

We scanned the DEX list for genes encoding protein methyltransferases, demethylases, acetyltransferases, deacetylases, kinases, phosphatases and E3 ligases. While we found a few that were up- or down-regulated in the 3D7+PfAlba3-Ty1 ring and trophozoite samples, for example genes encoding FIKK kinases, their role in Alba post-translational regulation remains to be established. The only known regulator, PfSir2, the histone deacetylase, is not misexpressed in the 3D7+PfAlba3-Ty1 strain.

6. In line 114: The functional validation of PfAlba6 has been reported (Nag et al.,2024) that is not mentioned.

Author response:

We have included this reference on Page 5, Lines 114-116.

Re: Spectrum00885-24R1 (Ectopic overexpression of *Plasmodium falciparum* DNA/RNA-binding Alba proteins misregulates virulence gene homeostasis during asexual blood development)

Dear Dr. Shruthi Sridhar Vembar:

Your manuscript has been accepted, and I am forwarding it to the ASM production staff for publication. Your paper will first be checked to make sure all elements meet the technical requirements. ASM staff will contact you if anything needs to be revised before copyediting and production can begin. Otherwise, you will be notified when your proofs are ready to be viewed.

Sincerely,
Björn Kafsack
Editor
Microbiology Spectrum

Reviewer #1 (Comments for the Author):

I thank the authors' for the responses and the revisions they have made to the manuscript, which have effectively addressed the concerns and suggestions raised during the review process.